# Healthy Eating and Active Lifestyles for Diabetes (HEAL-D): study protocol for the design and feasibility trial, with process evaluation, of a culturally tailored diabetes self-management programme for African-Caribbean communities

Louise M Goff,[1] Amanda P Moore,[1] Carol Rivas,[2] Seeromanie Harding[3]

¹Diabetes and Nutritional Sciences, Kings College London, London, UK
²Faculty of Health Sciences, University of Southamptom, Southamptom, UK
³Diabetes & Nutritional Sciences Division, King's College London, London, UK

**Correspondence to**
Dr Louise M Goff;
louise.goff@kcl.ac.uk

## ABSTRACT

**Introduction** Black British communities are disproportionately burdened by type 2 diabetes (T2D) and its complications. Tackling these inequalities is a priority for healthcare providers and patients. Culturally tailored diabetes education provides long-term benefits superior to standard care, but to date, such programmes have only been developed in the USA. The current programme of research aims to develop the Healthy Eating and Active Lifestyles for Diabetes (HEAL-D) culturally tailored T2D self-management programme for black British communities and to evaluate its delivery, acceptability and the feasibility of conducting a future effectiveness trial of HEAL-D.

**Methods and analysis** Informed by Medical Research Council Complex Interventions guidance, this research will rigorously develop and evaluate the implementation of the HEAL-D intervention to understand the feasibility of conducting a full-scale effectiveness trial. In phase 1, the intervention will be developed. The intervention curriculum will be based on existing evidence-based T2D guidelines for diet and lifestyle management; codesign methods will be used to foster community engagement, identify the intervention's underpinning theory, identify the optimal structure, format and delivery methods, ascertain adaptations that are needed to ensure cultural sensitivity and understand issues of implementation. In phase 2, the intervention will be delivered and compared with usual care in a feasibility trial. Process evaluation methods will evaluate the delivery and acceptability of HEAL-D. The effect size of potential primary outcomes, such as HbA1c and body weight, will be estimated. The feasibility of conducting a future effectiveness trial will also be evaluated, particularly feasibility of randomisation, recruitment, retention and contamination.

**Ethics and dissemination** This study is funded by a National Institute of Health Research Fellowship (CDF-2015-08-006) and approved by National Health Service Research Ethics Committee (17-LO-1954). Dissemination will be through national and international conferences, peer-reviewed publications and local and national clinical diabetes networks.

**Trial registration number** NCT03531177; Pre-results.

## Strengths and limitations of this study

► This study employs rigorous complex intervention methodology to develop and evaluate a culturally tailored diabetes self-management intervention.

► Participatory codesign methods are being used to foster stakeholder engagement in intervention development.

► The Capability, Opportunity, Motivation and Behaviour change framework is being used to identify appropriate intervention behaviour change techniques.

► Process evaluation measures are being collected to assess the feasibility of evaluating the intervention in a full-scale trial.

► The feasibility trial is designed to estimate the effect size of the intervention rather than efficacy, which will be the focus of a future definitive trial.

## INTRODUCTION

Type 2 diabetes (T2D) affects approximately 3 million people in England and consumes around 10% of the National Health Service (NHS) budget, estimated at almost £9 billion in 2011 and predicted to rise to 17% of the NHS budget by 2035.[1] Diabetes and its associated complications place an illness burden on patients and carers, which disproportionately affects those from ethnic minority backgrounds.[2] The estimated prevalence of T2D is up to three times higher for black British communities compared with white

Europeans.[3] T2D occurs, on average, 10 years earlier in black British people, the mean age of diagnosis is 48 years and approximately 25% of patients are under the age of 40 years.[4] Furthermore, glycaemic control is worse at the time of diagnosis and requires greater medical management, and poorer outcomes are evident.[5–7] The reasons for these disparities are not fully understood; while biological factors are involved, it is understood that a range of behavioural, lifestyle and health system factors play a role. Tackling these inequalities is a healthcare priority.[8 9]

Individuals of black British ethnicity form the second largest ethnic minority population in the UK; around 4% of the population self-identify from this ethnic background.[10] Around half of individuals are of black African ancestry and a third of black Caribbean ancestry.[10] Growth in the black British communities is relatively recent, beginning mainly in response post-Second World War appeals to citizens of the Commonwealth regions to assist with gaps in its labour market. This prompted a large influx of migrants in the 1950s from the Caribbean islands, particularly Jamaica. Migration from the African continent has been more recent, peaking around the 1980s; migrants from African nations currently form the largest growing ethnic minority group in the UK population.[11] In some regions, such as London, black British communities may represent 30%–40% of the local population and are therefore a 'majority-minority' community. Other demographic patterns are recognised; the age distribution of the black African and black Caribbean communities differs, with a larger proportion of black Caribbeans being aged 65 years and over, while in the black African population, a greater proportion are children and young adults. High rates of unemployment are evident, averaging around 12% compared with 4% in the white British population.[11]

Poor access to diabetes healthcare is a significant issue for minority ethnic groups.[2] In the UK, the NHS provides care to all UK residents that is free at the point of delivery. First-line diabetes management is situated in primary care and aims to promote patient involvement and self-management,[12] enabling patients to adopt a healthy lifestyle and to manage their diabetes through support and education.[13] To achieve this, UK T2D management guidelines recommend that all patients attend a structured education course to teach them the principals of T2D self-management and that this be offered annually from the time of diagnosis.[14] Courses are recommended to use a group structure; typically they use face-to-face delivery by a diabetes specialist nurse or dietitian, with lay educator codelivery in some cases.[14] Referral to such courses is audited and incentivised through the Quality Outcomes Framework.[15] Ethnic minority groups report finding it more difficult to access primary care services[16] and are more likely to report that they have not had the opportunity to attend a diabetes education course than white populations.[17] Specifically, African-Caribbean (AfC) communities often report a distrust of medical advice and a desire for natural, non-pharmacological therapies.[18]

Furthermore, healthcare professionals are perceived as lacking cultural understanding[19] and their advice as lacking cultural relevance[20] or being poorly adapted to culture and needs,[18] despite their intentions; these issues may contribute to the poorer diabetes outcomes and increased morbidity experienced by AfC patients.

Culturally tailored healthcare is proposed to be one of the main ways in which healthcare disparities can be addressed[21–23] and is identified as a priority by patients.[8] Culturally tailored diabetes education has demonstrated greater improvements in diabetes control and knowledge than usual care, and the benefits are maintained long term.[22 24] Culture is a concept that is notoriously difficult to define, but generally, within healthcare, it is thought of as 'a set of attitudes, values, beliefs and behaviours shared by a group of people, communicated from one generation to the next'.[25] In their model for understanding cultural sensitivity in healthcare, Resnicow et al[26] described two dimensions in culture: surface and deep structures. Tailoring interventions to surface structures involves matching materials and messages to observable, 'superficial' characteristics of a target population, for example, language and food, familiar to, and preferred by, the target audience. Deep structure involves incorporating the cultural, social, historical, environmental and psychological forces that influence the target health behaviours in the proposed target population. Whereas surface structure generally increases the 'receptivity' or 'acceptance' of messages, deep structure conveys salience.[26] Culture is ever evolving for any group, and it is important to recognise the diversity that exists within any one 'cultural group', which is particularly evident in migrant populations where second/third generations may have undergone significant acculturation. To date, culturally tailored interventions for the African diaspora have largely been based in the USA and may not translate to UK healthcare structures or UK AfC communities whose cultural needs may be different.[23]

A two-phase programme of research is proposed in which a culturally tailored, evidence-based self-management programme for T2D in African and Caribbean communities, called Healthy Eating & Active Lifestyles for Diabetes (HEAL-D), is developed, followed by a feasibility trial. The intervention curriculum will be based on existing evidence-based guidelines for T2D[14 27] to enable it to have potential to be embedded into clinical practice; codesign methods will be used to identify the optimal structure, format and methods of delivery and to ascertain appropriate adaptations that are needed to ensure cultural sensitivity of the content. The purpose of this article is to present the protocol for the development and feasibility trial of HEAL-D.

## Purpose and aims

The overall aims of this research are to develop a culturally tailored, evidence-based self-management programme for managing T2D among AfC communities in primary

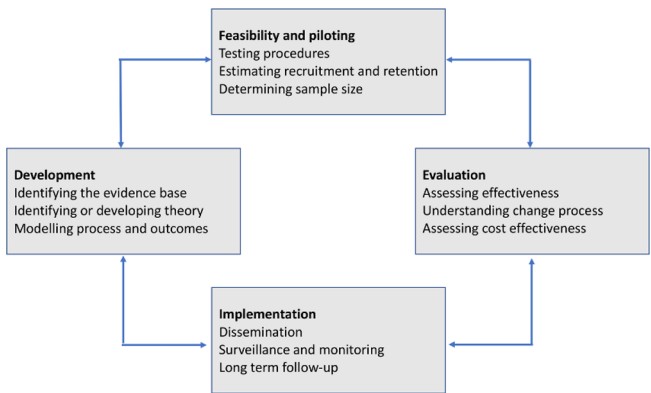

**Figure 1** Medical Research Council's framework for the development and evaluation of complex interventions. Reproduced from Craig *et al. British Medical Journal*. 2008; 337:a1655.

care, called HEAL-D, and to determine the feasibility of evaluating HEAL-D through a future effectiveness trial.

The objectives are to:

1. Develop a self-management programme, based on existing evidence-based diet and lifestyle guidelines, appropriately tailored for AfC patients through codesign methods.
2. Establish the feasibility of conducting an effectiveness trial of HEAL-D, considering issues such as participation rates and potential effect sizes.

## METHODS AND ANALYSIS

Guided by the Medical Research Council's Complex Interventions framework[28] (figure 1), two distinct phases of research are proposed: phase 1 is a formative phase in which the HEAL-D intervention will be developed; and phase 2 will evaluate the HEAL-D intervention in a feasibility trial. Study recruitment began in April 2017; the study duration will be 36 months.

### Phase 1: development of a culturally tailored T2D self-management programme

The process for the development of HEAL-D is outlined in figure 2. First, to ensure its potential to be embedded into future clinical practice, the HEAL-D curriculum will align with existing UK management recommendations and guidelines published by the National Institute of Clinical Excellence and Diabetes UK[14 27]:

Guidelines for diet and lifestyle management of T2D[27]:

1. Achieve 5%–10% wt loss or weight maintenance in those of healthy weight.
2. Undertake 150 min/week of moderate-to-vigorous intensity aerobic physical activity plus two sessions/week of strength training.
3. Balance carbohydrate intakes through portion control and promotion of low glycaemic index and wholegrain sources.
4. Limit saturated fat intake (<10% of energy intake), replace with monounsaturated fats.
5. Limit salt intake (<6 g per day).

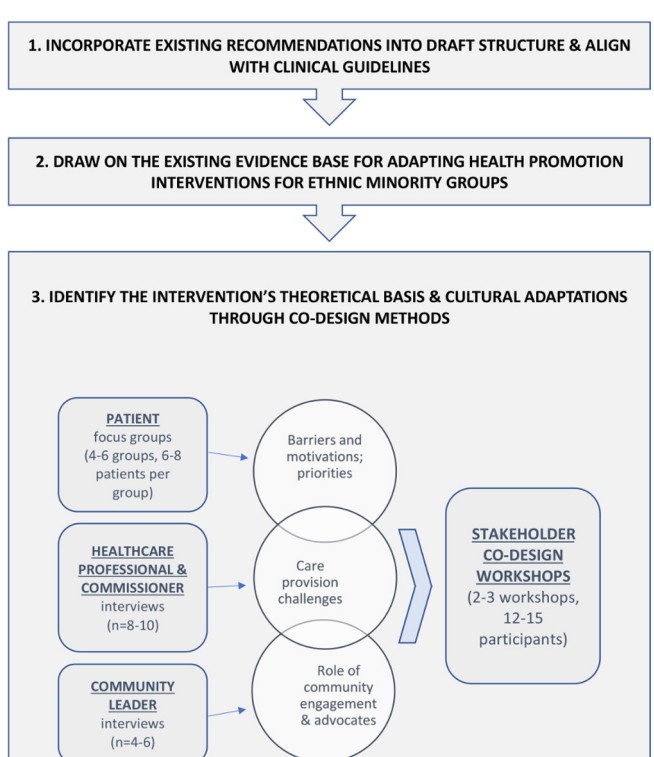

**Figure 2** Schematic diagram of phase I: development of HEAL-D using evidence synthesis and codesign methodology to design a culturally tailored self-management programme for T2D in African and Caribbean communities. HEAL-D, Healthy Eating & Active Lifestyles for Diabetes; T2D, type 2 diabetes.

6. Consume oily fish at least twice per week.

Guidelines for T2D recommend that self-management structured education is offered to adults with T2D and/or their family members or carers, with group education as the preferred option and that the education programmes are theory driven and evidence based and meet the cultural, linguistic, cognitive and literacy needs of the population.[14]

### Drawing on the existing evidence base

Second, it will draw on key themes reported in published literature relating to methodologies for adapting health promotion interventions for ethnic minority groups. These have been evaluated in a number of recent systematic reviews; aside from acknowledging the lack of UK-based studies, these reviews make several recommendations. The powerful influence of social networks on health beliefs and behaviours should be acknowledged,[29] and a focus on community-level interventions should be taken; delivering care in a social context promotes engagement and has been shown to be more effective than

traditional individual-centred behavioural approaches.[23] Community engagement should be promoted to overcome issues of deep-rooted, historical distrust of medical advice and settings, to develop and nurture trust between the researchers and community and to nurture the strong sense of collectivism and kinship networks that are evident among AfC communities. Participatory methods (eg, patient involvement in intervention design, lay-led delivery of interventions) should be employed as they are highly effective at improving health behaviours and self-efficacy across a number of conditions.[30] Using community gathering places (eg, faith institutions) as intervention settings offers the benefit of cultural relevancy and may reach populations who would not normally access self-management education.[31]

### Identifying the intervention's theoretical basis

Behavioural interventions should have a theoretical underpinning[28 32] so that the changes that are expected,

---

> ### Box 1 Topic guides for patient focus groups and stakeholder interviews
>
> **Patient focus groups**
> - ► Knowledge and perceptions of diabetes, and diet and lifestyle advice for managing diabetes.
> - ► Current practices relating to diabetes self-care, and diet and lifestyle.
> - ► Health concerns/priorities in relation to diabetes.
> - ► Motivations and barriers/difficulties relating to diabetes self-care, weight management and diet and lifestyle.
> - ► Experiences and perceptions of diabetes care/education and barriers to accessing care.
> - ► Experiences of behaviour change in relation to diabetes, weight, diet and lifestyle—successes and failures.
> - ► Role of family/friends/communities in influencing and shaping knowledge and behaviours in relation to diabetes, diet and lifestyle.
>
> **Community leader interviews (including religious leaders)**
> - ► Health problems affecting the community and diabetes impact on health within this context.
> - ► Attitude of the community towards health, medicines and doctors.
> - ► Role of community leaders in promoting health and community activities.
> - ► Diabetes health promotion activities within the community. What worked and what did not.
> - ► Barriers and facilitators to positive diabetes behaviours within the community.
> - ► Advice about engaging the community: who are the role models; what will engage and help people; how can healthcare and community work together.
>
> **Healthcare professional interviews**
> - ► Experience of supporting African and Caribbean patients. What are the issues. How could things be improved. What factors make successful T2D management likely.
> - ► Patient beliefs and motivations.
> - ► Involvement in community activities and experience of working with community leaders and lay educators and suggestions to improve partnerships.
> - ► Difficulties and challenges with offering a tailored lifestyle intervention.

---

and how these will be achieved, can be predicted from consideration of known behaviour change techniques. While there have been a number of interventions tailored to support diet and lifestyle behaviour change in AfC communities,[33] their theoretical underpinning has rarely been drawn out or clearly presented. The theoretical underpinning of HEAL-D will be developed through a combination of key themes from the published literature and new primary research.

In the literature, collectivism and the importance of social interaction for people of AfC ancestry is well reported,[29] and the provision of a social support group, or inclusion of a family member, has been shown to be particularly effective in lifestyle interventions in African-American communities.[34 35] These findings suggest social learning theory, which focuses on promoting behaviour change through social interaction, role modelling and social comparison, may be a relevant behaviour change theory for our intervention. Notably, much of literature that identifies the drivers of health behaviours in AfC communities and may, therefore, inform the theoretical basis of an intervention comes from the USA, and it is not known to what extent these findings apply to AfC in other regions. One of the reasons we will use codesign methods will be to understand the relevance of these existing themes to the UK context and enable us to identify themes that are important to black British communities.

### Codesigning the intervention through participatory methods

HEAL-D will use participatory codesign methods to engage patients, healthcare providers and community leaders (eg, church leaders and community group leads) in focus groups, interviews and workshops to achieve the following:
1. Foster community engagement.
2. Identify the theoretical underpinning of HEAL-D and its mechanisms of action.
3. Identify appropriate cultural adaptations for the intervention.
4. Understand issues of intervention implementation.

#### Focus groups and interviews

Focus groups, 8–10 groups of 6–8 participants, will be conducted with patients with T2D of AfC ethnicity, recruited through local churches, mosques and community groups, as well as through general practitioner (GP) practices in London. The focus groups will be conducted in local accessible community venues, for example, church hall, library and community centre. Patients will be purposively sampled to get a spread of socioeconomic position, generational status and ancestral origins, as principal factors impacting on health status, healthcare access and cultural behaviours in these groups.[36–38] Separate focus groups will be conducted with men and women, and patients of direct African versus Caribbean ancestry, as they report different cultural barriers/facilitators to lifestyle change.[36 37] A topic guide (box 1) based

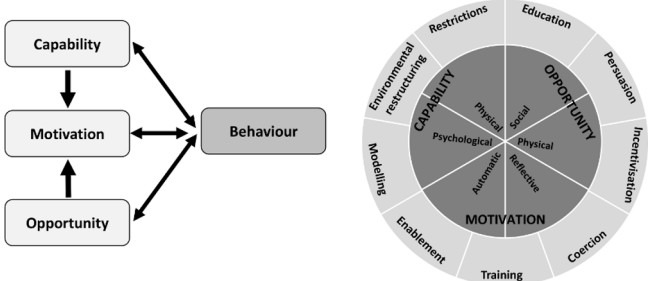

**Figure 3** The Capability, Opportunity, Motivation and Behaviour (COM-B) framework and behaviour change wheel; a framework for developing behavioural interventions. Reproduced from Michie S *et al*. *Implementation Science*. 2011; 6:42.

on themes identified in the literature will be used to steer discussions and ensure coverage of key themes while encouraging free discussion of opinion/perspective. Focus groups have been selected to enable us to understand normative needs, as suited to the development of a community intervention.

Semistructured interviews will be conducted with 8–10 healthcare providers, including general practitioners, practice nurses, diabetes specialist nurses, diabetes specialist dietitians and commissioners. The interviews will cover issues relating to healthcare needs and engagement of AfC patients, experiences of delivering healthcare to AfC patients and barriers and facilitators to working in partnership with community groups to deliver care for AfC communities (box 1). Interviews have been selected for this part of the study to enable us to gather a full range of experiences and therefore optimise implementation.

Community leaders representing faith and non-faith institutions (n=4–6) will be invited to participate in semi-structured interviews. Leaders will be identified initially through existing networks, for example, Diabetes UK Community Champions initiative. Word-of-mouth and

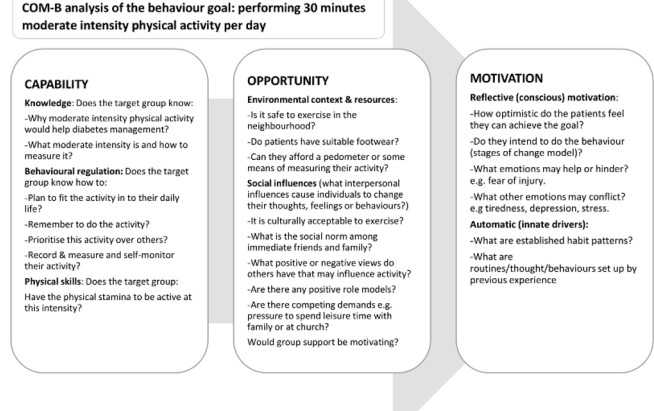

**Figure 4** Applying the COM-B behaviour change framework to the development of the HEAL-D intervention; identifying theory of change. COM-B, Capability, Opportunity, Motivation and Behaviour; HEAL-D, Healthy Eating and Active Lifestyles for Diabetes.

'snow-balling' techniques that are highly effective within these communities will be used to recruit a wider network. The interviews will cover issues relating to the role of community networks in promoting health of AfC communities, sustaining health among community members and opportunities for greater impact (box 1).

### Analysis

The focus groups and interviews will be digitally recorded and transcribed verbatim. The data will be analysed using the framework approach in NVivo (QSR International), theoretically driven by socioecological theory to identify themes relating to issues at the individual, family, community and healthcare delivery levels and how these influence self-efficacy and behaviour change. Our analysis will identify priority behaviours of focus for the intervention, key barriers and facilitators to behaviour change and healthcare engagement, favoured settings and a rudimentary draft of the cultural adaptations. Deviant case analysis, which is consideration of cases that do not fit the general picture, will be undertaken, though our primary interest is in the commonalities as this is a community-level intervention. Primary coding and development of a coding scheme will be carried out by a single researcher; a second researcher will independently use this coding scheme to code 20% of the data for cross-comparison to improve dependability. This will provide methodological rigour required for confidence in the analysis of the qualitative data. The themes will be fed back and discussed with a service user group (SUG), which will consist of representatives of patients, healthcare providers and community leaders. The SUG will be set up to inform and guide each stage of the research plan and will be a forum through which the research team can seek the opinion of key stakeholders, in this case particularly relating to interpretation of the qualitative data and to ensure trustworthiness of conceptualisations. The SUG will also review research documents, such as patient information sheets and questionnaires, and provide feedback on their content and suitability for the communities of focus.

We will divide our data into behavioural 'barriers' and 'facilitators' where possible. To ascertain appropriate behaviour change techniques for our intervention,[32] we will map our analysis onto the Capability, Opportunity, Motivation and Behaviour (COM-B) framework from the Behaviour Change Wheel[39] (figure 3), and then in each case, consider the outcome behaviours that our intervention will aim to achieve; a worked example is shown in figure 4. We will use the COM-B framework to identify appropriate *functions* of our intervention to optimise facilitators and overcome barriers to achievement of planned outcomes, for example, 'education' for capability barriers, 'modelling' for opportunity and motivation barriers. Finally, we will select specific behaviour change techniques, for example, education and goal setting, that focus on the specific functions we have identified. We will also look to identify other themes that arise from the data, which might not map clearly onto the COM-B framework

(eg, contextual themes relating to the health system) but which may inform our intervention theory as well as help us to understand issues of implementation (eg, favoured settings and timings). Through this analysis, we will identify our intervention theory that we will draw on for the next stage of the study, as documented through a logic diagram.

### Stakeholder codesign workshops

Following evaluation of the focus groups and interviews, our stakeholders, 12–15 patients, healthcare providers, commissioners and community leaders will be invited to participate in a series of 2–3 half-day workshops, held in community locations. The workshops will seek to gain stakeholder involvement in developing the details of the interventions. This will include determining the setting, the media channels, structure and delivery, as well as steering the research team to understand and respond to literacy and numeracy needs. The workshops will endeavour to reach a consensus opinion from attendees, but where stakeholders have different needs and a consensus cannot be reached, the research team will consult with the SUG to make decisions on the way forward and consider where there is scope for the intervention to be structured to meet these different needs, for example, delivery in a range of settings. In the first workshop, the research team will feed back the findings of the focus groups and interviews; anonymised interview extracts will be presented to illustrate the key themes and issues that were identified. The stakeholders will be asked to discuss the themes and behavioural targets in small groups, using directed tasks/questions to facilitate the discussions. Following the small group discussions, the researchers will facilitate discussion as a whole to clarify/confirm interpretation; open discussion/debate will be encouraged to examine the themes in depth and for all stakeholders to agree a mutual understanding.

In the second workshop, elements of the proposed intervention will be presented for comment, refining and development. Using scenarios, the stakeholders will be asked to brainstorm, in small groups, key issues relating to the scenarios. For example, the moderator will present scenarios relating to the intervention setting and the attendees will be asked to discuss and identify the pros and cons of each, and then feed back their discussions to the other attendees. The attendees will be asked to review existing educational/support materials, for example, leaflets and videos, and provide feedback on, for example, language/phrasing, content, pitch and understanding. The research team will then facilitate cross-discussion between groups to develop the conclusions and a consensus.

In the final workshop, draft intervention materials, developed from workshops 1 and 2, will be presented. For example, media channels that could be used to promote behaviour change such as testimonials, storytelling and cooking demonstrations. The stakeholders will be divided into small groups to discuss and provide feedback on the

acceptability of the components of the intervention and identify potential barriers to engagement. Following the small group discussions, the researchers will facilitate feedback and encourage discussion as a whole to clarify/confirm the researcher's interpretation. The intervention template may be further refined and will be developed into the detailed programme.

### Phase 2: evaluation of HEAL-D; a culturally tailored T2D self-management programme for African and Caribbean communities

In phase 2, a feasibility study, with an embedded process evaluation, will be conducted to address the following objectives:

1. Evaluate the HEAL-D intervention, particularly its theoretical under-pinning, acceptability, fidelity, issues of implementation and sustainability.
2. Evaluate the feasibility of trial procedures, considering issues such as rates of recruitment, retention, completion and contamination.
3. Estimate the effect size of potential trial outcomes including HbA1c, weight, waist circumference, blood pressure, dietary intake, physical activity levels, diabetes knowledge and quality of life, to inform an effectiveness trial.

### Study design

The feasibility study will use a randomised controlled design, with individual patients as the unit of randomisation, evaluating HEAL-D against usual care. In addition, there will be a cohort of phase 1 codesign patients who will be allocated to the intervention arm (not randomised) because their involvement in the intervention design phase would contaminate the control arm. These patients will be included in the feasibility study to enable us to evaluate the impact of former involvement on intervention engagement, acceptability and ownership.

### Participants

Participants will principally be recruited from general practice in the London Boroughs of Lambeth and Southwark through screening of referrals for structured education and letters of invitation to patients with established T2D. In addition, participants from the phase 1 codesign study will be invited to participate, and self-referral methods will also be used, for example, posters and advertisements in community locations.

Patients with diagnosed T2D who are of African or Caribbean ethnicity and with capacity to provide fully informed consent to participation in research will be eligible to participate in the trial. Ethnicity will be self-declared using the standard NHS ethnicity categorisation questionnaire. Patients who are unable to communicate in English and patients with complex therapeutic dietary needs may be ineligible to participate if their individual needs are deemed incompatible with the aims of the intervention. This is because the intervention will provide general diet and lifestyle advice for

the self-management of T2D in a group setting; in cases of patients with certain comorbidities, for example, advanced renal disease, the intervention may be inappropriate for the individual, and the group nature of the intervention will prevent their individual needs from being addressed.

A pragmatic sample size of 120 patients is anticipated to be sufficient to evaluate the programme, allowing for 20% dropout/non-completion; 80 patients will be randomised, 40 in each arm, and a further cohort of patients (n=40) from phase 1 will be allocated to the intervention arm without randomisation. As this is a feasibility trial, it will not be powered to detect statistically significant intervention effects. A primary objective of the study is to provide estimates of key parameters such as potential effect sizes, recruitment and retention rates of the trial and participation rates of the programme to enable the optimal design of a full-scale trial to be determined.

### Intervention and control arms
Participants in the control arm will continue with usual care deemed appropriate and delivered by their primary care team, which may include referral to group structured education and/or one-to-one consultations with healthcare professionals.

Participants in the intervention arm will be offered the HEAL-D programme, which will deliver a curriculum of culturally tailored, evidence-based diet and physical activity education and behaviour change in a group setting. In line with clinical guidelines, the programme will be delivered by trained educators (external to the research team); favoured educators (eg, lay educators vs healthcare professionals) will be identified in the codesign process. The details of each session, particularly the behaviour change techniques and corresponding activities/materials, will be identified through the codesign work.

The proposed curriculum will map to evidence-based guidelines and will be as follows:
1. An introduction to T2D self-management principles.
2. Physical activity in T2D management.
3. Carbohydrates and portion sizes.
4. Weight management for T2D.
5. Managing cardiovascular health.

In line with clinical guidelines for diabetes structured education, the education sessions will be delivered through educator-led interactive discussion; however, support materials will be provided to reinforce the learning, detailing evidence-based diet and physical activity guidance, which is culturally tailored for the African and Caribbean communities.

### Data collection
We will use a mixed methods approach, collecting a range of quantitative and qualitative data, to evaluate the intervention and the feasibility of trial procedures.

### Estimating the effect of the intervention on potential trial outcomes
Participants will attend a baseline and postintervention follow-up assessment visit, conducted by a research technician, at 26–32 weeks to collect the following potential trial outcomes and estimate effect sizes:

Biomedical outcomes: a 5 mL venous blood sample will be taken for analysis of HbA1c and total HDL-cholesterol and LDL-cholesterol and triglycerides. Systolic and diastolic blood pressure will be measured using an automated sphygmomanometer.

Anthropometric outcomes: body weight will be measured using digital scales, with the patient wearing light clothing (without shoes); height will be measured, using a stadiometer, without shoes; body mass index will be calculated as (weight [kg]/height [m$^2$]). Waist circumference will be measured with the patient wearing only light clothing, at the midpoint between the lowest rib and the iliac crest.

Diet and physical activity behaviour outcomes: dietary intake will be assessed through completion of a 24-hour diet recall, using the structured multiple pass interview method, and physical activity through 3-day Actiwatch accelerometer assessment and completion of the International Physical Activity Questionnaire.

The following validated self-complete questionnaires will be administered to assess: diabetes knowledge (Short Diabetes Knowledge Instrument[40]); diabetes and diet knowledge and competence (Perceived Diabetes & Dietary Competence[37]); empowerment (Diabetes Empowerment Scale-Short Form[41]); social support (Multidimensional Scale of Perceived Social Support[42]); diabetes distress (problem areas in diabetes (PAID)-5[43]); and quality of life (EuroQol (EQ)-5D-3L[44]).

Statistical analysis: given that this is a feasibility study with a small sample size, descriptive statistics will be used ($\chi^2$ test and Fisher's exact test). Differences between the groups in all outcomes will be estimated with 95% CIs. The descriptive data will provide stable estimates of the variability of continuous outcomes by group and provide estimates of differences between the groups in means and proportions for the key outcomes. The SD of the mean change in HbA1c will be estimated by arms and used to derive the sample size calculation for a subsequent trial.

### Evaluation of the HEAL-D intervention
Process evaluation is an essential part of testing complex interventions[45] and will be used in our feasibility trial to evaluate the HEAL-D intervention and the feasibility of trial procedures. Our process evaluation aims to achieve the following:
1. Test the intervention theory and whether the mechanisms of change operationalise as hypothesised.
2. Understand how the multiple components of the intervention interact.
3. Evaluate contextual factors that influence operationalisation of the intervention's theory/mechanisms of change and any unintended effects of these factors.

4. Evaluate whether the intervention is differentiable from 'usual practice'.

5. Evaluate implementation of the intervention, particularly 'reach' (eg, who receives the intervention), 'dose' and completion rates, and intervention fidelity (eg, coverage of core materials and learning objectives during delivery, and the extent to which the programme is delivered in accordance with the delivery manual, what adaptations are undertaken and why).

6. Evaluate acceptability of the intervention to patients, healthcare professionals and commissioners.

7. Evaluate intervention embedding and sustainability, for example, what are the barriers and facilitators to the uptake of the intervention in current care pathways.

A range of quantitative and qualitative data will be collected, as detailed in table 1. Attendance records, observation checklists, session/programme evaluation forms completed by patients and records of session activities completed by educators will provide quantitative data and will be used to evaluate a number of process domains, as indicated in table 1. Our process evaluation will mainly focus on qualitative evaluations, with which we will use inductive reasoning to determine whether the intervention requires further development and adaptation. Patient interviews and focus groups, and interviews with educators, healthcare professionals and commissioners, and session observation notes will provide qualitative data for the evaluation of a number of process domains, as detailed in table 1.

### Evaluation of trial procedures
The feasibility of trial procedures will be evaluated, particularly rates and methods of recruitment, retention, completion, contamination between study arms and the proposed data collection methods:

#### Recruitment
Several different pathways of recruitment will be implemented, for example, screening of primary care databases and letters of invitation, face-to-face referral during medical appointments, self-referral via posters and word-of-mouth referral. We will assess uptake rates from these different pathways to enable us to identify the most effective methods and assess the feasibility of recruiting for a full-scale trial.

#### Retention and completion
We will assess the rate of retention both within the HEAL-D intervention (ie, numbers completing each session and the full programme) and the feasibility trial (ie, numbers completing baseline and endpoint assessment visits). We will evaluate the feasibility of randomising and retaining a control arm by assessing dropout rates and comparing these between the study arms; we will also interview control arm patients to explore the acceptability of being assigned to the control arm.

#### Data collection methods
We will assess the frequency of missing data and any trends in which data is missing, for example, self-complete questionnaires, blood measures, to assess the feasibility of our data collection methods.

#### Contamination
We will interview patients from the control arm to explore issues of contamination, for example, did their participation in the trial promote change in self-management behaviours or motivate information-seeking behaviours, did they know anybody in the intervention arm or discuss the intervention with anybody.

### Patient and public involvement
Service user involvement is intrinsic to this proposed research, which uses participatory methods to engage patients and other stakeholders in the intervention design. The protocol provides extensive detail of how patients will be involved in the design, recruitment, conduct and dissemination of the research.

### Ethics and dissemination
All data will be anonymised and data protection protocols followed.

The study findings will be disseminated to the scientific community via conference presentations and peer-reviewed manuscripts and to healthcare professionals via national and local clinical networks. The findings of the study will be communicated to our participants and local communities via the community networks and figureheads who we will engage in our participatory methods; we will give presentations at church events and publish a newsletter via our study website (www.heal-d.co.uk).

### DISCUSSION
This paper presents the protocol for the design and feasibility testing of HEAL-D, a culturally tailored T2D self-management programme for UK African and Caribbean communities. This study will employ rigorous complex intervention methodology to develop and evaluate the implementation of a culturally tailored T2D self-management intervention. The intervention's curriculum will be based on existing evidence-based guidelines for diet and lifestyle management of T2D; participatory codesign methods will be employed to foster community engagement and partnership. We will use a 'bottom-up' approach to identify the cultural adaptations of our intervention and identify its theoretical basis through thematic analysis and the COM-B change framework. The feasibility study will provide us with key information about the feasibility of running a full-scale trial of HEAL-D, and process evaluation methods will enable us to understand how and why the intervention is effective or ineffective.

To date there have been no tailored education programmes for black British communities. Indeed, it is not known to what extent culturally tailored care is

**Table 1** Mapping of the HEAL-D feasibility study research questions, process evaluation data sources and evaluation methods

| Process evaluation domain and research questions | Data sources | | | | | | | | | | Evaluation method |
|---|---|---|---|---|---|---|---|---|---|---|---|
| | Patient questionnaires | Session observations | Session record of activities | Patient evaluation forms | Educator interviews | Patient interviews | Patient focus groups | Attendance records | HCP interviews | Commissioner interviews | |
| **Testing intervention theory and mechanisms of change** | | | | | | | | | | | |
| Are the intervention's mechanisms of change operationalised as hypothesised? | X | X | X | X | X | X | X | | | | Qualitative data collected through interviews/focus groups with patients and educators, and session observation notes will be used to evaluate how the theory of the intervention operationalises and interacts with contextual factors. |
| How is the operationalisation of the mechanisms of change influenced by contextual factors? | | X | X | X | X | X | X | | | | |
| Does the interaction of the mechanisms of change with contextual factors give rise to unintended effects? | | X | X | | X | X | X | | | | |
| **Assessing usual practice and contamination** | | | | | | | | | | | |
| Is HEAL-D differentiable from 'usual practice'? | | | | | | X | | | | | Interviews will be conducted with patients from both arms. Experiences of the intervention and control will be explored. With control patients issues of contamination and perceptions of 'usual care' will be discussed. |
| Is there contamination in control patients? | | | | | | X | | | | | |
| **Assessing implementation** | | | | | | | | | | | |
| What is the intervention reach and dose? | X | | | | | | | X | | | Questionnaire data will assess who receives the intervention and how representative they are, for example, age, gender, ethnicity and working status. Attendance records will be used to quantify the proportion of patients receiving the full versus part intervention. |
| Are the HEAL-D components/sessions delivered with fidelity and what is the nature of any adaptions? | | X | X | | X | | | | | | To assess fidelity and compare intervention deliveries and contextual impacts educators will complete a record of activities and materials and list any resources/activities/discussions that were additional to the standardised schedule. These will be explored in depth in educator interviews that will be conducted at the end of the programme delivery. The research team will observe HEAL-D delivery to quantitatively assess coverage of curriculum, use of supporting materials and behaviour change techniques, quality of delivery and participant engagement (binary score or a 5-point Likert scale). Observers will qualitatively document course adaptations and general contextual observations. |
| Does the delivery of HEAL-D differ between sites, and what gives rise to differences? | | X | X | | X | | | | | | |
| How well are the HEAL-D components/sessions delivered? | | X | X | | | | | | | | |
| **Assessing intervention acceptability** | | | | | | | | | | | |

Continued

**Table 1** Continued

| Process evaluation domain and research questions | Data sources | | | | | | | | | | | Evaluation method |
| | Patient questionnaires | Session observations | Session record of activities | Patient evaluation forms | Educator interviews | Patient interviews | Patient focus groups | Attendance records | HCP interviews | Commissioner interviews | |
|---|---|---|---|---|---|---|---|---|---|---|---|
| Is HEAL-D acceptable to patients, commissioners and healthcare professionals? | | X | | X | | X | X | | X | X | Acceptability will be evaluated through a range of qualitative and quantitative data. Quantitative data will be generated in patient evaluations, which will use 10-point scales to assess their views on the quality of the programme content, structure, format and delivery; the sessions/programme will be deemed 'acceptable' where they score ≥6 points. Interviews/focus groups with patients, educators, healthcare professionals and commissioners will explore acceptability through qualitative data, for example, reasons for attendance/non-attendance among patients and suggestions for amendments. |
| **Assessing intervention sustainability** | | | | | | | | | | | |
| How likely is the HEAL-D intervention to be sustainable and what factors might ensure sustainability? | | | | | | | | | X | X | Qualitative data collected through interviews with healthcare professionals and commissioners will be used to evaluate barriers and facilitators to implementation of HEAL-D into current care pathways, and its fit with organisational priorities, and the feasibility of sustained resource allocation to the HEAL-D intervention if found to be successful. |

HCP, healthcare professionals; HEAL-D, Healthy Eating & Active Lifestyles for Diabetes.

needed for black British communities as little work has been undertaken with these communities. Our codesign work is intended to explore the sociocultural barriers and facilitators to behaviour change and structure HEAL-D accordingly. We acknowledge that we are likely to find huge diversity within our black British communities and *culture* will likely be only one of many important factors that affects their health behaviours. However, our codesign work will provide a more comprehensive theoretical underpinning for the content of our programme than that which currently exists and will provide us with a framework on which to evaluate the effectiveness of our programme. This work will provide essential information and evaluation to inform the design of a future definitive trial.

**Contributors** All authors have made substantial contributions to this study. LMG, CR and SH were responsible for the conception and design of the study. All authors developed the protocol and study approach. LMG drafted the manuscript. All authors read, revised and approved the final manuscript. LMG is guarantor.

**Funding** This report is independent research arising from a Career Development Fellowship (LMG, CDF-2015-08-006) supported by the National Institute for Health Research.

**Disclaimer** The views expressed in this publication are those of the author(s) and not necessarily those of the NHS, the National Institute for Health Research or the Department of Health.

**Competing interests** None declared.

**Patient consent for publication** Not required.

**Ethics approval** The study protocol has been approved by the Health Research Authority (London Fulham Research Ethics Committee; 17/LO/1954); all participants will provide written consent prior to participation.

**Provenance and peer review** Not commissioned; externally peer reviewed.

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
