## [Reviewer comments · BMJ Open]

This paper was submitted to a another journal from BMJ but declined for publication following peer review. The authors addressed the reviewers' comments and submitted the revised paper to BMJ Open. The paper was subsequently accepted for publication at BMJ Open.

(This paper received three reviews from its previous journal but only two reviewers agreed to published their review.)

ARTICLE DETAILS

TITLE (PROVISIONAL)	Healthy Eating and Active Lifestyles for Diabetes (HEAL-D): study protocol for the design and feasibility trial, with process evaluation, of a culturally-tailored diabetes self-management programme for African-Caribbean communities
AUTHORS	Goff, Louise; Moore, Amanda; Rivas, Carol; Harding, Seeromanie

VERSION 1 – REVIEW

REVIEWER	Cornelia Straßner University Hospital Heidelberg, Department of General Practice and Health Services Research, Germany
REVIEW RETURNED	27-Sep-2018

GENERAL COMMENTS	This is a very comprehensive study protocol with a very complex approach of intervention development. Some passages need clarifications: Abstract: • Information about the planned RCT is lacking.• See also my comment on purpose and aim. It might be necessary to make clear that the core curriculum of the HEAL-D-programme already exists and that the content will be tailored (if I understood correctly).• Strength and limitations: Limitations of the study are not at all mentioned in the whole paper. Introduction: • The authors describe that the prevalence of T2D is 3 times higher in Black-British communities with poorer outcomes. It would be interesting to know if anything is known whether this is mainly due to worse health behaviour and worse access to health care (the target of the study) or which role the genetic risk plays.• The aim of the authors is to develop a cultural-tailored intervention. It would be worth to reflect on the definition and role of culture in a few sentences. Developing a culturally-
--

tailored intervention implies to a certain degree that patients of African-Caribbean origin are regarded as a homogenous group which is distinct from the homogenous group of the average population. There is huge body of literature which criticizes concepts of culture based on nationality/ancestry and emphasizes the importance of understanding diversity within one “culture”. Numerous other factors might be more important for health-related behaviours than “culture”. This might be especially important if a migrant already lives for a long time or in the second/third generation in the other country (it seems that this is not at all considered in the study).

I think the paper would be more balanced if the authors stated that they are aware of this debate and describe how the intervention will be tailored if it turns out during the development phase that cultural issues are not a major barrier.

- Some epidemiological data to describe Afro-Caribbean communities in England would be interesting. What’s their proportion of the general population, how many are in first/second/third generation in England, education etc.

Purpose and aims:

- It is not really clear to me which is the intervention to be evaluated and which parts of the intervention are already set and where is room for tailoring. It took me a while to understand that it is already defined that the Heal-D-programme will consist of an educational session in a group setting with an already elaborated curriculum (page 15/16). The aim of the study is to culturally tailor the content of this curriculum? If this is correct I would make this clear already in the introduction section and describe the evidence-base of this pre-set curriculum. Will other interventions, e.g. to engage key stakeholders – be part of the Heal-D-programme and thus of the evaluation or will they be engaged only during the development phase?

Methods and analysis

- The part about phase 1 is very long and difficult to follow. It could be shortened or structured with more headlines to make it better readable.
- Line 131-133: There is an undefined reference in brackets (REF NICE 2014). I think the sentence is grammatically not correct.
- Line 316: It would be interesting to know what is “usual care” for diabetes in England? Are there for example Disease Management Programmes?
- Line 331: Who will lead the educational sessions? Members of the study team?

	 Line 338: Who will conduct the assessment, the study team? The planned start date of the intervention period should be stated Sample size calculation: The authors did not perform a sample size calculation since the study is planned as a feasibility study. Yet it would be interesting to know which effect size is needed to detect group differences with 120 patients. Tables and figures:  Figure 2 is a table and should be named as such Figure 3 is rather a box or a table
--	---

REVIEWER	Jessica L. McCurley, PhD MPH Massachusetts General Hospital / Harvard Medical School, USA
REVIEW RETURNED	09-Oct-2018

GENERAL COMMENTS	Summary: This manuscript summarizes the authors' plans for (1) the development of, and (2) a feasibility trial and process evaluation of, a culturally-tailored diabetes self-management program targeting Afro-Caribbean individuals with type 2 diabetes. This represents an important body of work that may result in substantial benefit for a community experiencing disparities of type 2 diabetes incidence and control. Major Concerns: 1. My most pressing concern is that this manuscript primarily describes the planned methodology for the development process of an intervention, rather than the evaluation of an intervention. The authors have included tentative plans for intervention implementation and evaluation, but many aspects of the intervention content and evaluation are yet to be determined. Thus, I find the current title and abstract of the manuscript to be inconsistent with its contents, and both should be modified to describe the manuscript more appropriately. I am unsure whether it is common practice for BMJ Open to publish manuscripts describing the methodology of an intervention trial before the intervention has been fully developed. Further, while the intervention development process described appears to be rigorous and well-founded (and could be of utility to other researchers), I am not sure of the utility to readers of a methods paper describing a feasibility trial and process evaluation before core elements of the intervention (e.g., setting, length, content, cultural tailoring procedures) have been established. Basically, I am unsure whether the second half of this paper is ready for publication at the current time. It makes sense to me for authors to complete the intervention development process, and THEN publish a methods paper which describes the development process but can also describe in more detail the intervention itself and plans for evaluation. I will defer to the editors regarding whether this issue is problematic enough to reject the paper or recommend that the authors to publish the development methods separately. Because I strongly believe in the potential and importance of the work the authors are undertaking, I do hope there is a way to successfully publish a revised version of this manuscript in BMJ-Open.
--

2. Authors state, "The theoretical basis for HEAL-D will be identified through two processes; firstly an evidence synthesis of key themes through published literature relating to adapting health promotion interventions for ethnic minority groups, and secondly through new primary research." The former of these two seems to be something that the authors can and should complete now, to include in the publication of this article. Authors continue on page 8 to cite some important methodological themes and findings from prior research, but these are not theories.

3. Relatedly, while I absolutely understand the important of the co-design process for determining optimal intervention implementation methods, it is not clear to me how the authors expect the co-design procedures to expand the theoretical basis on which the intervention will be developed.

4. The process evaluation section needs to be carefully revised for clarity and detail. Throughout this section, authors are not specific about how their determinations of evaluated domains (e.g., high versus low acceptability or feasibility) will be made. As is, this section remains quite vague. Typically, in a methods paper you want to give enough information do that other researchers could replicate what you have done (or are planning to do).

5. Authors need to find a way integrate Tables 1-3 and improve their readability. All three tables provide related information about the process evaluation, but how they are separated is confusing and not intuitive. It would more clear and efficient and less repetitive to see the planned process evaluation (domains, data source, evaluation method) in one table. Further, please include concise but specific information about how each element will be evaluated. Currently authors list data sources but (as mentioned above) authors do not say how determinations of high or low acceptability, for example, will be determined. Finally, please label tables appropriately with numbers and title (or as specified by the journal), and clarify why there appears to be repeated content in the main manuscript document and the supplementary tables.

Minor points:

1. For your citation of the percentage of the NHS budget and amount dedicated to T2D (in the introduction), is there a more updated estimate than 2011? I imagine costs have increased since then.

2. Be careful with your verb tenses and be consistent throughout (e.g., our intervention "is designed" vs. "will be designed")

3. Do African-Caribbean individuals exhibit worse diabetes control only at the time of diagnosis? Or in general/over time? If in general, you might state that instead.

4. It would be helpful in your introduction to include further background and demographics for readers regarding the African-Caribbean community – how many African-Caribbean individuals are there currently/what percentage of the UK population, whether this a growing demographic, whether they experience other health disparities or major structural barriers, etc. Also please be specific about the younger age at which T2D is diagnosed... at what age is first diagnosis common?

	5. Please spell out any acronyms before they are used (such as “MRC” Complex interventions framework) – and it would be helpful to explain this framework as well, or show a visual representation in a figure. 6. It does not appear that all in-text references have been appropriately formatted (see p. 7 line 132). 7. Please explain what a Service User Group is or is meant to be. 8. It would be helpful to show the COM-B Framework and Behavior Change Wheel in a figure, if possible. 9. Authors alternate between the term “enablers” and “facilitators”– are these synonyms? Please make it clear if they are synonyms, or if not, what is the distinction. 10. This sentence was unclear to me: “Themes that do not map clearly onto the COM-B framework will also inform the programme theory e.g. contextual themes at the community and health system levels.” Please explain further. 11. How will you handle disagreements/differences in opinion in the co-design workshops, if participants do not manage to achieve consensus? Will stakeholders have the time and resources to be able to attend 3 half-day workshops? 12. It appears that there is an incomplete sentence in line 311. 13. Did authors consider an attention control instead of usual care? Is it possible that intervention effects may in part be related to increased attention and group social support? 14. Are there any planned adaptations for low literacy or numeracy, if these are concerns in this population? 15. Have the proposed self-report measures been validated in Afro-Caribbean populations? If not, have you considered including them in the co-design workshops, for review of cultural relevance and comprehensibility by community members? 16. This sentence was particularly unclear to me: “Sustainability will be considered by assessing the scope for the intervention to be embedded within current care pathways, and contextual factors that may influence decision-making around continuance.” (p. 18). Please clarify and be specific about what you mean by scope and continuance – and how an assessment regarding sustainability will actually be determined.
--	--

VERSION 1 – AUTHOR RESPONSE

REVIEWER 1.

This represents an important body of work that may result in substantial benefit for a community experiencing disparities of type 2 diabetes incidence and control.

RESPONSE: We thank the reviewer for their support for our work.

Major Concerns:

1. My most pressing concern is that this manuscript primarily describes the planned methodology for the development process of an intervention, rather than the evaluation of an

intervention. The authors have included tentative plans for intervention implementation and evaluation, but many aspects of the intervention content and evaluation are yet to be determined. Thus, I find the current title and abstract of the manuscript to be inconsistent with its contents, and both should be modified to describe the manuscript more appropriately.

RESPONSE We apologise for this. We have amended the manuscript title and abstract to include more reference to the development work. Due to our focus on communities that are normally neglected, there is a lot of emphasis on our development work, however, we have ensured that the section describing our feasibility trial includes all the measures that are recommended for feasibility trials (NIHR 2017).

2. I am unsure whether it is common practice for BMJ Open to publish manuscripts describing the methodology of an intervention trial before the intervention has been fully developed. Further, while the intervention development process described appears to be rigorous and well-founded (and could be of utility to other researchers), I am not sure of the utility to readers of a methods paper describing a feasibility trial and process evaluation before core elements of the intervention (e.g., setting, length, content, cultural tailoring procedures) have been established. Basically, I am unsure whether the second half of this paper is ready for publication at the current time. It makes sense to me for authors to complete the intervention development process, and THEN publish a methods paper which describes the development process but can also describe in more detail the intervention itself and plans for evaluation. I will defer to the editors regarding whether this issue is problematic enough to reject the paper or recommend that the authors to publish the development methods separately. Because I strongly believe in the potential and importance of the work the authors are undertaking, I do hope there is a way to successfully publish a revised version of this manuscript in BMJ-Open.

RESPONSE Thank you for your support for our work, we are delighted that you recognise the potential and importance of our study. We aim to publish this protocol to describe the entirety of our planned work; we plan to subsequently publish an intervention development paper that details the findings of our co-design work.

3. Authors state, "The theoretical basis for HEAL-D will be identified through two processes; firstly an evidence synthesis of key themes through published literature relating to adapting health promotion interventions for ethnic minority groups, and secondly through new primary research." The former of these two seems to be something that the authors can and should complete now, to include in the publication of this article. Authors continue on page 8 to cite some important methodological themes and findings from prior research, but these are not theories.

RESPONSE Thank you for highlighting this, we agree with your comment. We have amended this section accordingly: in lines 179-189 we describe methodologies from the literature, and in lines 198-201 we have described theories which have been identified in the literature and which will inform our intervention.

4. Relatedly, while I absolutely understand the important of the co-design process for determining optimal intervention implementation methods, it is not clear to me how the authors expect the co-design procedures to expand the theoretical basis on which the intervention will be developed.

RESPONSE We apologise if this is not clear. We have added information in lines 193-195 & 203-208 to try and clarify that our co-design work will help us to explore whether the themes from the literature are relevant to the UK context.

5. The process evaluation section needs to be carefully revised for clarity and detail. Throughout this section, authors are not specific about how their determinations of evaluated domains (e.g., high versus low acceptability or feasibility) will be made. As is, this section remains quite vague. Typically, in a methods paper you want to give enough information do that other researchers could replicate what you have done (or are planning to do).

RESPONSE Thank you for this comment. We have made extensive revision to the process evaluation section, pages 18-20, to improve the clarity and detail. We have provided more detail of our methods in a revised table (now Table 2).

6. Authors need to find a way integrate Tables 1-3 and improve their readability. All three tables provide related information about the process evaluation, but how they are separated is confusing and not intuitive. It would more clear and efficient and less repetitive to see the planned process evaluation (domains, data source, evaluation method) in one table. Further, please include concise but specific information about how each element will be evaluated. Currently authors list data sources but (as mentioned above) authors do not say how determinations of high or low acceptability, for example, will be determined. Finally, please label tables appropriately with numbers and title (or as specified by the journal), and clarify why there appears to be repeated content in the main manuscript document and the supplementary tables.

RESPONSE We have revised our tables and integrated the tables into one process evaluation table (Table 2). We have provided more detail about our data sources and evaluation methods, and ensured there is no repetition or duplication between the main manuscript and the tables. We have provided more detail about how we will evaluate our quantitative data e.g. intervention acceptability, however our data will be largely qualitative and we will be interested mainly in understanding whether further intervention adaptations are needed rather than dichotomising these domains.

Minor points:

1. For your citation of the percentage of the NHS budget and amount dedicated to T2D (in the introduction), is there a more updated estimate than 2011? I imagine costs have increased since then.

RESPONSE We have expanded this part of the introduction, line 63; unfortunately there is not a robust updated estimate of actual costs available but we have cited a forecast for the increasing costs.

2. Be careful with your verb tenses and be consistent throughout (e.g., our intervention “is designed” vs. “will be designed”)

RESPONSE Apologies, these have been corrected throughout.

3. Do African-Caribbean individuals exhibit worse diabetes control only at the time of diagnosis? Or in general/over time? If in general, you might state that instead.

RESPONSE We have expanded this part of the introduction to show that greater management is needed also, line 69.

4. It would be helpful in your introduction to include further background and demographics for readers regarding the African-Caribbean community – how many African-Caribbean individuals are there currently/what percentage of the UK population, whether this a growing demographic, whether they experience other health disparities or major structural barriers, etc. Also please be specific about the younger age at which T2D is diagnosed... at what age is first diagnosis common?

RESPONSE Thank you for raising this, we have added in extra information to provide background to the UK context, lines 67-68, 73-87.

5. Please spell out any acronyms before they are used (such as “MRC” Complex interventions framework) – and it would be helpful to explain this framework as well, or show a visual representation in a figure.

RESPONSE We have ensured all acronyms are now spelt out in full and added a figure of the MRC framework, figure 1.

6. It does not appear that all in-text references have been appropriately formatted (see p. 7 line 132).

RESPONSE Thank you for pointing this out, this has been corrected.

7. Please explain what a Service User Group is or is meant to be.

RESPONSE We have added detail on this, lines 260-265.

8. It would be helpful to show the COM-B Framework and Behavior Change Wheel in a figure, if possible.

RESPONSE We have added a figure to show the COM-B framework and Behaviour Change Wheel, figure 3.

9. Authors alternate between the term “enablers” and “facilitators”– are these synonyms? Please make it clear if they are synonyms, or if not, what is the distinction.

RESPONSE We have corrected our terminology to ensure we use ‘facilitators’ throughout.

10. This sentence was unclear to me: “Themes that do not map clearly onto the COM-B framework will also inform the programme theory e.g. contextual themes at the community and health system levels.” Please explain further.

RESPONSE We apologise if this was not clear. We have expanded our description, lines 276-280, to provide more detail.

11. How will you handle disagreements/differences in opinion in the co-design workshops, if participants do not manage to achieve consensus? Will stakeholders have the time and resources to be able to attend 3 half-day workshops?

RESPONSE We have added detail to this section, lines 287-291.

12. It appears that there is an incomplete sentence in line 311.

RESPONSE Apologies, this has been amended.

13. Did authors consider an attention control instead of usual care? Is it possible that intervention effects may in part be related to increased attention and group social support?

RESPONSE Thank you for raising this interesting suggestion. We gave much consideration to our control, particularly whether it should be usual care or a standard education programme. The literature suggests that group social support may be important for the communities we are working with so we expect this to be part of the theoretical basis of our intervention. This would make it difficult to have an attention control.

14. Are there any planned adaptations for low literacy or numeracy, if these are concerns in this population?

RESPONSE We have added some detail to cover this point, line 287.

15. Have the proposed self-report measures been validated in Afro-Caribbean populations? If not, have you considered including them in the co-design workshops, for review of cultural relevance and comprehensibility by community members?

RESPONSE Thank you for raising this important point. There are a lack of validated questionnaires for the communities we are working with and one of the aims of our work will be to look at how well the questionnaires work with our participants. One of the roles of our service user group is to give us feedback on the methods/tools we are proposing to use. We have added this to the manuscript, lines 263-265.

16. This sentence was particularly unclear to me: “Sustainability will be considered by assessing the scope for the intervention to be embedded within current care pathways, and contextual factors that may influence decision-making around continuance.” (p. 18). Please clarify and be specific about

what you mean by scope and continuance – and how an assessment regarding sustainability will actually be determined.

RESPONSE This section has been rewritten in response to earlier comments.

REVIEWER 2

This is a very comprehensive study protocol with a very complex approach of intervention development.

RESPONSE We thank the reviewer for their support for our work.

Some passages need clarifications:

Abstract:

- Information about the planned RCT is lacking.

RESPONSE We have indicated that the feasibility study will inform the design of a future RCT but at this stage we cannot provide any further detail.

- See also my comment on purpose and aim. It might be necessary to make clear that the core curriculum of the HEAL-D-programme already exists and that the content will be tailored (if I understood correctly).

RESPONSE Thank you for raising this. We have amended the abstract to include these details.

- Strength and limitations: Limitations of the study are not at all mentioned in the whole paper.

RESPONSE Thank you for this recommendation. We have added a discussion to the end of the paper to detail our perceived strengths and limitations.

Introduction:

- The authors describe that the prevalence of T2D is 3 times higher in Black-British communities with poorer outcomes. It would be interesting to know if anything is known whether this is mainly due to worse health behaviour and worse access to health care (the target of the study) or which role the genetic risk plays.

RESPONSE We have added this detail to the introduction, lines 70-72.

- The aim of the authors is to develop a cultural-tailored intervention. It would be worth to reflect on the definition and role of culture in a few sentences. Developing a culturally tailored intervention implies to a certain degree that patients of African-Caribbean origin are regarded as a homogenous group which is distinct from the homogenous group of the average population. There is huge body of literature which criticizes concepts of culture based on nationality/ancestry and emphasizes the importance of understanding diversity within one “culture”. Numerous other factors might be more important for health-related behaviours than “culture”. This might be especially important if a migrant already lives for a long time or in the second/third generation in the other country (it seems that this is not at all considered in the study).

I think the paper would be more balanced if the authors stated that they are aware of this debate and describe how the intervention will be tailored if it turns out during the development phase that cultural issues are not a major barrier.

RESPONSE We thank the reviewer for this important point, which we agree with. We have tried to draw on this in both the introduction, lines 109-122, and the discussion, lines 496-502.

- Some epidemiological data to describe Afro-Caribbean communities in England would be interesting. What's their proportion of the general population, how many are in first/second/third generation in England, education etc.

RESPONSE We have added more detail to the introduction, lines 73-87.

Purpose and aims:

- It is not really clear to me which is the intervention to be evaluated and which parts of the intervention are already set and where is room for tailoring. It took me a while to understand that it is already defined that the Heal-D-programme will consist of an educational session in a group setting with an already elaborated curriculum (page 15/16). The aim of the study is to culturally tailor the content of this curriculum? If this is correct I would make this clear already in the introduction section and describe the evidence-base of this pre-set curriculum.

Will other interventions, e.g. to engage key stakeholders – be part of the Heal-D-programme and thus of the evaluation or will they be engaged only during the development phase?

RESPONSE We apologise for our lack of clarity. We have amended the manuscript to make this point clearer, lines 129-132, 141-142, 153-170.

Methods and analysis

- The part about phase 1 is very long and difficult to follow. It could be shortened or structured with more headlines to make it better readable.

RESPONSE This part has been revised and some information tabulated to make it more succinct and easier to follow.

- Line 131-133: There is an undefined reference in brackets (REF NICE 2014). I think the sentence is grammatically not correct.

RESPONSE Apologies, this has been corrected now.

- Line 316: It would be interesting to know what is “usual care” for diabetes in England? Are there for example Disease Management Programmes?

RESPONSE We have added these detail to the introduction, lines 92-97.

- Line 331: Who will lead the educational sessions? Members of the study team?

- Line 338: Who will conduct the assessment, the study team?

RESPONSE We have added these details, lines 365-367, 385.

- The planned start date of the intervention period should be stated

RESPONSE This has been added to the manuscript, line 150.

- Sample size calculation: The authors did not perform a sample size calculation since the study is planned as a feasibility study. Yet it would be interesting to know which effect size is needed to detect group differences with 120 patients.

RESPONSE Thank you for this point. Unfortunately we have little data upon which to base a sample size calculation as very few studies to date have engaged UK African-Caribbean patients so we don't know if they are comparable to the general population in terms of HbA1c, body weight etc.

Tables and figures:

- Figure 2 is a table and should be named as such
- Figure 3 is rather a box or a table

RESPONSE These have been revised and corrected.

VERSION 2 – REVIEW

REVIEWER	Cornelia Straßner University Hospital Heidelberg Department of General Practice and Health Services Research
REVIEW RETURNED	12-Nov-2018

GENERAL COMMENTS	The authors have been responsive to most of my comments. The introduction improved substantially and provides the necessary information to understand the background of the trial. The method section still appears somewhat unbalanced: There is an extensive description of the development phase and the process evaluation of the intervention with many details on the planned qualitative procedures while the effectiveness analysis is described rather briefly. The quantitative outcomes are not at all mentioned in the abstract. There is no section “statistical analysis” describing how the quantitative outcomes will be analyzed. Furthermore, the authors state that recruitment has already begun half a year ago in April 2018 (the sentence says: recruitment WILL BEGIN in April 2018, but it should say HAS BEGUN to make clear to the readers that this is a retrospective protocol). Recruitment of whom exactly? It would be interesting to know how far the intervention development process is now. I agree with reviewer 1 that it is doubtful whether the extensive descriptions of the planned methods for intervention development without actually describing the final intervention is profitable for the reader especially if the intervention development is already in an advanced stage. Maybe it would be better to split the paper into two: A paper describing the development and final design of the intervention and a paper describing the feasibility trial with process and effectiveness analysis. Finally it is the decision of the editors if the general content of the paper is acceptable for the journal. Minor comments:  • References of the validated questionnaires should be provided • Line 198: While there have been a number of interventions References of these interventions should be provided. • The information provided in line 492 – 506 is already described in the introduction and method section and does not have to be repeated in the discussion section. Generally it should be checked if redundancies can be avoided.
--

REVIEWER	Jessica McCurley, PHD MPH Massachusetts General Hospital/Harvard Medical School
REVIEW RETURNED	02-Dec-2018

GENERAL COMMENTS	This revised manuscript appears much improved, with careful attention paid to most reviewer comments from the previous review. A few additional clarifications and improvements are needed.  1. Further precision and clarity is needed regarding the name “HEAL-D.” In some places in the manuscript, this is referred to as a “program of research” that involves multiple steps and aims (e.g., all of the processes described in this manuscript), while in other places it refers to the intervention to be developed. Even after multiple readings, I remain quite confused about what “HEAL-D” is meant to be. Please clarify and be consistent with the use of this critical phrase. 2. Line 154 refers to “existing management recommendations and guidelines” but does not state where these guidelines come from. Please inform readers of the origins of these guidelines in the text. 3. In regard to the section titled, “Identifying the interventions’ theoretical basis,” I do not think that authors have sufficiently addressed my original concerns about the inclusion of previous literature on behavior change theory. The new text in this section
--

	presents information about intervention elements (e.g., the inclusion of family members) that have been associated with efficacy – but this is not behavioral change theory. What should be cited in this section are specific pre-existing behavior change theories that authors believe might apply to their intervention. Relatedly, I continue to be confused about exactly how authors plan to use the co-design method to generate new theories. Perhaps it is more appropriate to talk about identifying possible mechanisms of action, or barriers and facilitators, in the co-design process - versus generating new theory? 4. The use of bullet point lists and numbered lists throughout this manuscript gives it an informal and incomplete feel. Further, it is unclear why some lists are numbered and others have only bullet points. Is this in line with the journal's style recommendations? It is my recommendation that authors minimize the use of lists and be consistent with formatting if any lists are used. 5. Line 151 contains a spelling error in the word "programme." Line 443 includes a wrong word ("effectiveness" should be "effective.") Please check the manuscript carefully for spelling and other clerical errors.
--	---

VERSION 2 – AUTHOR RESPONSE

Reviewer: 1

Method section: there is an extensive description of the development phase and process evaluation while the effectiveness analysis is described rather briefly. The quantitative outcomes are not at all mentioned in the abstract.

RESPONSE: The work we are undertaking is largely qualitative. Our trial is a true feasibility trial that focuses heavily on understanding process and issues of intervention implementation e.g. fidelity, contamination. Our quantitative data are intended to enable us to estimate the effect size of our intervention, rather than an assessment of efficacy. We have added some more information to the abstract regarding the quantitative measures (lines 33-35) but we are cautious not to describe an efficacy trial.

There is no "statistical analysis" section describing how the quantitative outcomes will be analyzed.

RESPONSE: We have added in a statistical analysis section (lines 402-408) to describe our handling of the quantitative outcomes, which will be largely descriptive analyses, in line with our feasibility methods.

The authors state that recruitment WILL BEGIN, but it should say HAS BEGUN to make clear that this is a retrospective protocol.

RESPONSE: We apologise, this inaccuracy has arisen because the manuscript was originally submitted to the journal prior to commencing recruitment. This has now been corrected (line 150).

Maybe it would be better to split the paper into two: a paper describing the development and final design of the intervention and a paper describing the feasibility trial with process and effectiveness analysis.

RESPONSE: The Editor has advised that the journal agrees to us publishing our whole protocol as a single manuscript.

- References of the validated questionnaires should be provided

RESPONSE: These have been added (lines 396-400).

- Line 198: references of these interventions should be provided.

RESPONSE: These have been added (line 191).

• The information provided in line 492 – 506 is already described and does not have to be repeated in the discussion section.

RESPONSE: The discussion has been cut back to avoid repetition.

Reviewer: 2

1. Further precision and clarity is needed regarding the name “HEAL-D.” In some places it is referred to as a “program of research” while in others it refers to the intervention to be developed.

RESPONSE: We apologise for the lack of clarity. Changes have been made throughout the manuscript to make it clear that HEAL-D is the intervention (lines 18-19; 31-32; 126-129; 148-149).

2. Line 154 refers to “existing management recommendations and guidelines”. Please inform readers of the origins of these guidelines in the text.

RESPONSE: Alongside the references we previously provided we have now added the publishers of these guidelines to clarify (lines 154-155).

3. “Identifying the interventions’ theoretical basis”. What should be cited in this section are specific pre-existing behavior change theories that authors believe might apply to their intervention. Relatedly, I continue to be confused about exactly how authors plan to use the co-design method to generate new theories. Perhaps it is more appropriate to talk about identifying possible mechanisms of action, or barriers and facilitators, in the co-design process - versus generating new theory?

RESPONSE: Intervention theory is a complex area with varying definitions in different fields, including those of health psychology/behaviour change, and those of implementation science in which the focus is on implementation/process theory, which is often more ecologically grounded. Our work is novel and will draw on both behaviour change and implementation theory. There are many theories that align with our approach, for example social learning theory and the socio-ecological model; our formative work is designed to enable us to take a comprehensive approach to theory building/enhancing. We have added reference to social learning theory (lines 198-200) as a pre-existing behaviour change theory that we think may apply to our intervention, and we have added more detail to the aims of our co-design workshops (lines 211). We have reviewed our manuscript carefully to ensure that we have not referred to ‘generating new theory’; our approach is to ‘identify’ relevant theories by which our intervention will bring about change, which will draw not only on behaviour change theory but also implementation theory.

4. It is my recommendation that authors minimize the use of bullets/lists.

RESPONSE: Several the bulleted lists have been expanded into paragraphs (lines 166-169; 174-186; 384-400), leaving only numbered bullets that refer to aims/objectives or specific lists.

5. Line 151 contains a spelling error in the word “programme.” Line 443 includes a wrong word (“effectiveness” should be “effective.”)

RESPONSE: Apologies, these have been corrected and the manuscript checked for errors.